# On the Optimal Error Exponent of Type-Based Distributed Hypothesis Testing [note 1]

**DOI:** 10.3390/e25101434

**Published:** 2023-10-10

**Authors:** Xinyi Tong, Xiangxiang Xu, Shao-Lun Huang

**Affiliations:** 1Tsinghua–Berkeley Shenzhen Institute, Shenzhen 518055, China; txy18@mails.tsinghua.edu.cn; 2Tsinghua Shenzhen International Graduate School, Shenzhen 518055, China; xuxx@mit.edu

**Keywords:** hypothesis testing, distributed system, information theory, local geometry

## Abstract

Distributed hypothesis testing (DHT) has emerged as a significant research area, but the information-theoretic optimality of coding strategies is often typically hard to address. This paper studies the DHT problems under the type-based setting, which is requested from the popular federated learning methods. Specifically, two communication models are considered: (i) DHT problem over noiseless channels, where each node observes i.i.d. samples and sends a one-dimensional statistic of observed samples to the decision center for decision making; and (ii) DHT problem over AWGN channels, where the distributed nodes are restricted to transmit functions of the empirical distributions of the observed data sequences due to practical computational constraints. For both of these problems, we present the optimal error exponent by providing both the achievability and converse results. In addition, we offer corresponding coding strategies and decision rules. Our results not only offer coding guidance for distributed systems, but also have the potential to be applied to more complex problems, enhancing the understanding and application of DHT in various domains.

## 1. Introduction

Distributed hypothesis testing (DHT) is a significant problem in the field of information theory [1]. In this problem, each distributed node observes partial data generated from the joint distribution and transmits an encoded message through a communication channel to a decision center, aiming to detect the true hypothesis. The primary goal of DHT is to maximize the decision error exponent in the asymptotic regime, where many different communication models [2,3,4,5,6] were considered in the previous literature. The main challenges of the DHT arise in two respects. Firstly, due to the intricate distributed structures, most of the existing works have focused on demonstrating achievability results, with converse results being limited to specific cases, such as the 1-bit [3], $\log_{2}3$-bit [7], and $O(\log_{2}n)$-bit [1] communication channels. Secondly, many of the achievability results were established using random coding with auxiliary random variables [8], which are difficult to implemen in real systems.

Notice that the distributed encoders in many real applications are required to process high-dimensional data [9], such as images, texts, and audios. Consequently, many of the federated learning algorithms focus on computing the quantities, such as the statistics, empirical risks, and gradient of data [10], which can be viewed as certain functions of the empirical distribution (type) of the data (for example, given the data $x_{1},\ldots ,x_{n}$ and feature function f(x), the statistic $\frac{1}{n}\sum_{i=1}^{n}f\left(x_{i}\right)=\sum_{x}{\hat{P}}_{X}\left(x\right)f\left(x\right)$ is a linear function of the empirical distribution ${\hat{P}}_{X}$).

Motivated by this observation, we investigate the optimal decision error exponent of DHT based on the empirical distributions (type-based) under two common communication models. The first problem considers a noiseless channel, which is the typical mathematical model in real federated learning scenarios. It comes from the reality that federated learning often assumes that the nodes and the center machine can exchange information precisely; however, the dimensionalities of the transmitted signals are limited [9]. Specifically, it is assumed that each node can only transmit the empirical mean of a one-dimensional feature, and such settings have gained significant attention recently in federated and multi-modal machine learning [9,11]. The second problem assumes that the signal of each node, encoded with the empirical distribution, is transmitted over an additive white Gaussian noise (AWGN) channel, which is a widely-used mathematical model for real-world channels [12]. The main goal of this paper is to establish the optimal error exponent for the aforementioned two problems by presenting: (i) the converse bound for the error exponent; and (ii) a practical coding strategy that achieves the converse bound.

The contributions of this paper are summarized as follows. First, in Section 4.1, we demonstrate the optimal error exponent for the type-based hypothesis testing over noiseless channels, where one-dimensional functions for all nodes and the corresponding decision rule are provided. Moreover, by applying the information geometric approach in [13], the hypotheses and the feature functions of each node can be modeled as vectors in the joint and marginal distribution spaces, respectively. In Section 4.3, the optimal feature function of each node can be interpreted as a decomposition of the hypothesis vector in the joint distribution space into vectors in the marginal distribution spaces, where each decomposed component indicates the contribution of the corresponding node in making the inference.

Second, we establish the optimal achievable error of the type-based hypothesis testing over AWGN channels by presenting both the achievability and converse results. In particular, the achievability part is based on a mixture coding strategy of both the amplify-and-forward and decode-and-forward strategies. Specifically, when the observed empirical distribution at a distributed node is sufficiently close to one of the true marginal distributions with respect to the two hypotheses, the node is confident of the true hypothesis. Then, we apply the decode-and-forward strategy, which first estimates the true hypothesis based on the observed empirical distribution, and then we apply the binary phase shift keying (BPSK) to transmit the decoded bit to the decision center. On the other hand, when the observed empirical distribution is far from both true marginal distributions, we apply the amplify-and-forward strategy to encode and transmit the observed empirical distribution by the pulse amplitude modulation (PAM) to the decision center. By applying the proposed coding strategy and conducting the log-likelihood ratio test at the decision center, we show in Section 5.2 the achievable error exponent. Finally, we demonstrate the converse results of the error exponent in Section 5.3 based on a genie-aided approach. The main idea is to add additional information to the distributed nodes. By either leveraging the true hypothesis to the distributed nodes or eliminating the channel noises, we show that the error exponent in Section 5.2 is also an upper bound of the optimal error exponent, which establishes the optimality.

## 2. Problem Formulations

Suppose that there are *K* random variables $X^{K}≜\left(X_{1},\ldots ,X_{K}\right)$. In this paper, we consider the binary hypothesis testing problem, and the two hypotheses $H_{0}$ and $H_{1}$ are defined as:(1)$$\begin{array}{cc} \; & H_{0}:\left(x_{1}^{\left(1\right)},\cdots ,x_{K}^{\left(1\right)}\right),\cdots ,\left(x_{1}^{\left(n\right)},\cdots ,x_{K}^{\left(n\right)}\right)\overset{i.i.d.}{∼}P_{X^{K}}^{\left(0\right)},\\ \; & H_{1}:\left(x_{1}^{\left(1\right)},\cdots ,x_{K}^{\left(1\right)}\right),\cdots ,\left(x_{1}^{\left(n\right)},\cdots ,x_{K}^{\left(n\right)}\right)\overset{i.i.d.}{∼}P_{X^{K}}^{\left(1\right)}, \end{array}$$
where the observable data are i.i.d. generated according to either $P_{X^{K}}^{\left(0\right)}$ or $P_{X^{K}}^{\left(1\right)}$ from the alphabet set $\left(X_{1},\cdots ,X_{K}\right)$. In addition, we assume that there are *K* distributed nodes, where the *k*-th (k=1,⋯,K) node can only observe the samples $X_{k}≜\left\{x_{k}^{\left(1\right)},\ldots ,x_{k}^{\left(n\right)}\right\}$. To facilitate clarity in our illustration, we concentrate on the discrete case, assuming that each alphabet $X_{k}$ is discrete, and $X≜X_{1}\times \cdots \times X_{K}$. In addition, for a joint distribution $Q_{X^{K}}\in P_{X}$, we use ${\left[Q_{X^{K}}\right]}_{X_{k}}$ to denote its marginal distribution with respect to $X_{k}$. We also denote $P_{X_{1}}^{\left(i\right)},\cdots ,P_{X_{K}}^{\left(i\right)}$ as the marginal distributions of $P_{X^{K}}^{\left(i\right)}$, for i=0,1. In the distributed hypothesis testing problem, we introduce a common assumption in the distributed setup [14] that the generating distributions $P_{X^{K}}^{\left(0\right)}$ and $P_{X^{K}}^{\left(1\right)}$ satisfy $D\left(P_{X^{K}}^{\left(1\right)}∥P_{X^{K}}^{\left(0\right)}\right)<\infty ,D\left(P_{X^{K}}^{\left(0\right)}∥P_{X^{K}}^{\left(1\right)}\right)<\infty$, to avoid the trivial irregularities. Due to the type-based restriction, we further assume that $P_{X_{k}}^{\left(0\right)}\neq P_{X_{k}}^{\left(1\right)}$, k=1,⋯,K. Otherwise, the transmitted message as a function of the empirical distribution would be uninformative for distinguishing the hypotheses. In the following, we denote ${\hat{P}}_{X_{k}}$ as the empirical distributions of $X_{k}$, defined as:(2)$$\begin{array}{c} {\hat{P}}_{X_{k}}\left(x_{k}\right)≜\frac{1}{n}\sum_{i=1}^{n}1\left\{x_{k}=x_{k}^{\left(i\right)}\right\}. \end{array}$$

### 2.1. Type-Based Hypothesis Testing over Noiseless Channels

As shown in Figure 1, node *k* (k=1,⋯,K) can encode the observed data $X_{k}$ and transmit a scalar signal by function $u_{k}$. Due to the computational requirement as introduced in Section 1, we impose a restriction whereby the encoder $u_{k}$ is explicitly dependent on the empirical distribution ${\hat{P}}_{X_{k}}$, i.e., $u_{k}:P_{X_{k}}\mapsto R$, and $P_{X_{k}}$ denotes the set of probability distributions defined on the alphabet $X_{k}$. For the most direct method, we can transmit the emprical distributions by encoding them into the real space, which can lead to computational difficulty for federated learning data. In this paper, we further consider one of the most commonly used approaches in federated learning [15,16] and assume that $u_{k}$ computes a one-dimensional statistic
(3)$$\begin{array}{c} u_{k}\left({\hat{P}}_{X_{k}}\right)=\frac{1}{n}\sum_{i=1}^{n}f_{k}\left(x_{k}^{\left(i\right)}\right)=E_{{\hat{P}}_{X_{k}}}\left[f_{k}\left(X_{k}\right)\right], \end{array}$$
where feature function $f_{k}:X_{k}\mapsto R$. Then, the decision center collects statistics ${\left\{u_{k}\left({\hat{P}}_{X_{k}}\right)\right\}}_{k=1}^{K}$, and makes a decision $\hat{H}$ on the true hypothesis. We prove in Section 4 that the further restrictions of computing the empirical means of features are without a loss of generality, where we can make good decisions as we observe the types. Additionally, the error probability is defined as
$$\begin{array}{c} P_{n}\left(\hat{H}\neq H\right)≜\sum_{i\in \{0,1\}}P_{H}\left(H_{i}\right)P_{n}\left(\hat{H}\neq H|H=H_{i}\right), \end{array}$$
where H denotes the true hypothesis, $P_{H}\left(H_{0}\right)$ and $P_{H}\left(H_{1}\right)$ are the prior distributions, and $P_{n}\left(\cdot \right)$ is the probability measure defined from the data sampling process (Equation 1). In particular, we focus on the asymptotic error decaying rate, i.e., the error exponent, defined as
(4)$$\begin{array}{c} E≜\lim_{n\to \infty }-\frac{1}{n}\log P_{n}\left(\hat{H}\neq H\right), \end{array}$$
where all logarithms are base *e* unless otherwise specified. The goal is to find the maximal error exponent of (Equation 4) and design the feature functions $f_{1},\cdots ,f_{k}$ and the detailed decision rule such that this error exponent can be achieved based on the log-likelihood ratio test (LLRT).

### 2.2. Type-Based Hypothesis Testing over AWGN Channels

As depicted in Figure 2, we employ the identical hypothesis testing formulation as presented in (Equation 1). In this context, it is assumed that nodes 1 through *K* encode and transmit a length-*m* sequence using functions $g_{1},\cdots ,g_{K}$, which operate based on their respective observations through additive white Gaussian noise (AWGN) channels to the central decision center. To accommodate the computational constraints, we restrict that the encoder $g_{k}$ (k=1,⋯,K) is a function of the empirical distribution ${\hat{P}}_{X_{k}}$, i.e.,
(5)$$\begin{array}{c} g_{k}:P_{X_{k}}\mapsto R^{m},\;k=1,\cdots ,K. \end{array}$$
Moreover, the averaged power constraints of the AWGN channels are:(6)$$\begin{array}{c} \frac{1}{m}E\left[{∥g_{k}\left({\hat{P}}_{X_{k}}\right)∥}^{2}\right]\leq p_{k},\;k=1,\cdots ,K, \end{array}$$
where the expectations are taken over the data sampling process defined in (Equation 1). Then, the decision center makes a decision $\hat{H}$ based on the received signals $g_{1}\left({\hat{P}}_{X_{1}}\right)+Z_{1},\cdots ,g_{K}\left({\hat{P}}_{X_{K}}\right)+Z_{K}$, where the noises are drawn from
(7)$$\begin{array}{c} Z_{k}∼N\left(0,\sigma_{k}^{2}I_{m}\right),\;k=1,\cdots ,K, \end{array}$$
and $I_{m}$ denotes the m×m identity matrix.

Additionally, we make the following assumption to make the errors arising from the AWGN channels and the decision process comparable, so that the trade-off between them can be described. In detail, we assume that the sequence length *m* also increases with *n*, and there exists a positive constant μ such that
(8)$$\begin{array}{c} \lim_{n\to \infty }\frac{n}{m(n)}=\mu . \end{array}$$

Our goal is to design the optimal encoders $g_{1},\cdots ,g_{K}$, subjected to the constraints (Equation 5) and (Equation 6), as well as the decision rule $\hat{H}$, where we have assumed $P_{H}\left(H_{0}\right)=P_{H}\left(H_{1}\right)=\frac{1}{2}$ for explicit mathematical expression, such that the error exponent as defined in (Equation 4) is maximized.

## 3. Related Works

Distributed hypothesis testing problems, also known as multiterminal hypothesis testing [1,3,14] or decentralized detection [17,18], have been extensively explored in the literature. In scenarios where each node can observe a single observation and send an encoded message to the central machine, the authors of [17] demonstrated that determining the optimal coding scheme is NP-hard, while [18,19] provided characterizations for the minimum decoding error rate and the optimal coding scheme for conditionally independent nodes.

Furthermore, in situations where each node can observe *n* samples and transmit an encoded message to the decision center, [3,5,14,20] investigated the optimal decoding error exponents for the case of K=2 nodes, with [21] generalizing the results to K>2 nodes. Additionally, the author of [5] studied the Neyman–Pearson-like test, which further constrained the encoded messages to being an empirical functional mean, and provided optimal functions for the scenario with K=2 nodes. The outcome presented in Section 4 can be perceived as a generalization of such setups to the case with K>2 nodes.

On the other hand, DHT over noisy channels represents a novel and highly significant sub-problem within the broader context. While current research has primarily focused on transmission over discrete memoryless channels, certain aspects of this sub-problem have been investigated. For instance, some studies have explored scenarios involving side information [22] and cases that counteract independence assumptions [23]. Additionally, optimal Type-II error considerations have been examined [24], along with investigations into the optimal pairs of Type-I and Type-II errors [25].

Diverging from the existing literature, the present paper delves into the DHT problem in the context of widely considered AWGN channels while also addressing the implications of computational demands. This novel approach fills a critical research gap and extends the understanding of DHT to a broader set of channel conditions, thus contributing to the advancement of the field.

## 4. Type-Based Hypothesis Testing over Noiseless Channels

In this section, we present the optimal error exponent along with the corresponding decision rule for the type-based hypothesis testing over noiseless channels. We commence by introducing the optimal error exponent under the condition that the decision center has access to the empirical distributions from different nodes.

**Definition** **1.**
*The quantities $D_{i}^{*}\left(R_{X_{1}},\cdots ,R_{X_{K}}\right)$, for i=0,1, are defined as*

(9)
$$\begin{array}{c} D_{i}^{*}\left(R_{X_{1}},\cdots ,R_{X_{K}}\right)≜\min_{Q_{X^{K}}\in S}\;D\left(Q_{X^{K}}∥P_{X^{K}}^{\left(i\right)}\right), \end{array}$$

*where*

$$\begin{array}{cc} S≜ & \left\{Q_{X^{K}}:{\left[Q_{X^{K}}\right]}_{X_{k}}=R_{X_{k}},\;k=1,\cdots ,K\right\}, \end{array}$$

*which represents the set of all distributions with given marginals $R_{X_{1}},\cdots ,R_{X_{K}}$.*


The following result provides the operational meaning of (Equation 9), which can be proved by Sanov’s theorem [12].

**Lemma** **1.**
*When $H_{i}$ is the true hypothesis, the probability that nodes 1,⋯,K observe the empirical distributions ${\hat{P}}_{X_{1}},\cdots ,{\hat{P}}_{X_{K}}$, respectively, is given by*

$$\begin{array}{c} P_{n}\left({\hat{P}}_{X_{1}},\cdots ,{\hat{P}}_{X_{K}}|H=H_{i}\right)≐\exp\left(-nD_{i}^{*}\left({\hat{P}}_{X_{1}},\cdots ,{\hat{P}}_{X_{K}}\right)\right),\;i=0,1, \end{array}$$

*where ≐ is the conventional dot-equal notation, i.e., we denote $f_{n}≐g_{n}$ when $\lim_{n\to \infty }\frac{1}{n}\log f_{n}=\lim_{n\to \infty }\frac{1}{n}\log g_{n}$. In addition, by applying the log-likelihood ratio test to detect the true hypothesis, the optimal decision error exponent based on the empirical distributions is*

(10)
$$\begin{array}{c} E^{*}≜\min_{R_{X_{1}},\cdots ,R_{X_{K}}}\;\max_{i\in \{0,1\}}\;D_{i}^{*}\left(R_{X_{1}},\cdots ,R_{X_{K}}\right). \end{array}$$



Note that the type-based hypothesis testing problem assumes that the signal from each node is a function of the empirical distribution. Hence, the optimal error exponent in (Equation 4) will not exceed $E^{*}$. In the following, we prove that error exponent $E^{*}$ can be achieved and provide the corresponding decision rule.

### 4.1. Optimal Feature

First, we introduce the following definitions of exponential and linear families, which will be useful for delineating our results.

**Definition** **2**(Exponential family)**.**
*Given distribution $P_{Z}\left(z\right)$, and a function T:Z→R, we define the distribution ${\tilde{P}}_{Z}^{\left(\lambda \right)}\left(\;\cdot \;;T,P_{Z}\right)$ as*
(11)$$\begin{array}{c} {\tilde{P}}_{Z}^{\left(\lambda \right)}\left(z;T,P_{Z}\right)≜P_{Z}\left(z\right)\exp\left(\lambda T\left(z\right)-\alpha \left(\lambda \right)\right),\;\mathrm{for}\;\mathrm{all}\;z\in Z, \end{array}$$
*with $\alpha \left(\lambda \right)≜\log\sum_{z^{\prime }\in Z}P_{Z}\left(z^{\prime }\right)\exp\left(\lambda T\left(z^{\prime }\right)\right)$. In addition, we use*
(12)$$\begin{array}{c} E_{Z}\left(T,P_{Z}\right)≜\left\{{\tilde{P}}_{Z}^{\left(\lambda \right)}\left(\;\cdot \;;T,P_{Z}\right):\lambda \in R\right\} \end{array}$$
*to denote the exponential family passing through $P_{Z}$ with T being the natural statistic.*

**Definition** **3**(Linear family)**.**
*Given a function h:Z→R, we define the linear family $L_{Z}\left(h\right)$ as*
(13)$$\begin{array}{c} L_{Z}\left(h\right)≜\left\{Q_{Z}\in P^{Z}:E_{Q_{Z}}\left[h(Z)\right]=0\right\}. \end{array}$$
*In addition, we define the half-spaces $S_{Z}^{\left(0\right)}\left(h\right)$ and $S_{Z}^{\left(1\right)}\left(h\right)$ as*
$$\begin{array}{c} S_{Z}^{\left(0\right)}\left(h\right)≜\left\{Q_{Z}\in P^{Z}:E_{Q_{Z}}\left[h(Z)\right]\leq 0\right\},\\ S_{Z}^{\left(1\right)}\left(h\right)≜\left\{Q_{Z}\in P^{Z}:E_{Q_{Z}}\left[h(Z)\right]\geq 0\right\}. \end{array}$$

Then, for i=0,1 and t>0, we define the sets
$$\begin{array}{cc} D_{i}\left(t\right) & ≜\{\left(R_{X_{1}},\cdots ,R_{X_{K}}\right):D_{i}^{*}\left(R_{X_{1}},\cdots ,R_{X_{K}}\right)<t\}. \end{array}$$
We also define $D\left(t\right)≜D_{0}\left(t\right)\cap D_{1}\left(t\right)$. It can be verified that, for all t≥0, both $D_{0}\left(t\right)$ and $D_{1}\left(t\right)$ are convex subsets of $P_{X_{1}}\times \cdots \times P_{X_{K}}$, and thus D(t) is also convex. In addition, we have the following lemma.

**Lemma** **2.**
*For $E^{*}$ as defined in *(10)*, we have D(t)=⌀ for all $t\in [0,E^{*}]$ and D(t)≠⌀ for all $t>E^{*}$. Additionally, a unique $\left({\tilde{R}}_{X_{1}},\cdots ,{\tilde{R}}_{X_{K}}\right)\in P_{X_{1}}\times \cdots \times P_{X_{K}}$ exists such that*

(14)
$$\begin{array}{c} D_{0}^{*}\left({\tilde{R}}_{X_{1}},\cdots ,{\tilde{R}}_{X_{K}}\right)=D_{1}^{*}\left({\tilde{R}}_{X_{1}},\cdots ,{\tilde{R}}_{X_{K}}\right)=E^{*}. \end{array}$$



**Proof.** See Appendix A. □

Based on Lemma 2, it follows from the separating hyperplane theorem (see, e.g., Section 2.5.1 of [26]) that functions $\left(f_{1}^{*},\cdots ,f_{K}^{*}\right)$, where $f_{k}^{*}:X_{k}\to R$, k=1,⋯,K exist, such that for all $\left(R_{X_{1}},\cdots ,R_{X_{K}}\right)\in D_{0}\left(E^{*}\right)$,
(15)$$\begin{array}{c} \sum_{i=1}^{K}\sum_{x_{i}\in X_{i}}R_{X_{i}}\left(x_{i}\right)f_{i}^{*}\left(x_{i}\right)=\sum_{i=1}^{K}E_{R_{X_{i}}}\left[f_{i}^{*}\left(X_{i}\right)\right]\leq 0, \end{array}$$
and for all $\left(R_{X_{1}},\cdots ,R_{X_{K}}\right)\in D_{1}\left(E^{*}\right)$,
(16)$$\begin{array}{c} \sum_{i=1}^{K}E_{R_{X_{i}}}\left[f_{i}^{*}\left(X_{i}\right)\right]\geq 0. \end{array}$$
Furthermore, we denote
(17)$$\begin{array}{c} h^{*}\left(x^{K}\right)≜\sum_{i=1}^{K}f_{i}^{*}\left(x_{i}\right), \end{array}$$
and then we have the following proposition. Given $P_{Z}\in P_{Z}$ and $S\subset P_{Z}$, we adopt the notation [27,28] $D\left(S∥P_{Z}\right)≜\inf_{Q_{Z}\in S}D\left(Q_{Z}∥P_{Z}\right)$, where $P_{Z}$ denotes the set of all distributions supported on Z.

**Proposition** **1.**
*The optimal exponent $E^{*}$ as defined in *(10)* satisfies*

(18)
$$\begin{array}{c} E^{*}=D\left(S_{X}^{\left(0\right)}\left(h^{*}\right)∥P_{X^{K}}^{\left(1\right)}\right)=D\left(S_{X}^{\left(1\right)}\left(h^{*}\right)∥P_{X^{K}}^{\left(0\right)}\right). \end{array}$$



**Proof.** See Appendix B. □

Consequently, we establish the optimality of $E^{*}$ and provide the corresponding decision rule.

**Theorem** **1.**
*Let $f_{1}^{*},\cdots ,f_{K}^{*}$ denote the features as defined in *(15)* and *(16)*. The optimal error exponent of *(4)* is given by*

(19)
$$\begin{array}{c} \lim_{n\to \infty }-\frac{1}{n}\log P_{n}\left(\hat{H}\neq H\right)=E^{*}, \end{array}$$

*where $E^{*}$ is defined in *(10)*. In addition, the corresponding decision rule $\hat{H}$ is*

(20)
$$\begin{array}{c} \sum_{k=1}^{K}E_{{\hat{P}}_{X_{k}}}\left[f_{k}^{*}\left(X_{k}\right)\right]\overset{\hat{H}=H_{1}}{\underset{\hat{H}=H_{0}}{≷}}0. \end{array}$$



**Proof.** See Appendix C. □

### 4.2. General Geometric Structure

The geometry associated with Proposition 1 and Theorem 1 is depicted in Figure 3. In this figure, each point represents a distribution in $P_{X}$, and the decision boundary (Equation 20) corresponds to the linear family $L_{X}\left(h^{*}\right)$ defined as in (Equation 13). In addition, from Corollary 3.1 of [27], $\lambda_{0},\lambda_{1}\in R$ exist such that
(21)$$\begin{array}{c} Q_{X^{K}}^{\left(i\right)}≜{\tilde{P}}_{X^{K}}^{\left(\lambda_{i}\right)}\left(\;\cdot \;;h^{*},P_{X^{K}}^{\left(i\right)}\right),\;i=0,1, \end{array}$$
satisfy
(22)$$\begin{array}{c} D\left(S_{X}^{\left(1-i\right)}\left(h^{*}\right)∥P_{X^{K}}^{\left(i\right)}\right)=D\left(Q_{X^{K}}^{\left(i\right)}∥P_{X^{K}}^{\left(i\right)}\right), \end{array}$$
where ${\tilde{P}}_{X^{K}}^{\left(\lambda_{i}\right)}\left(\;\cdot \;;h^{*},P_{X^{K}}^{\left(i\right)}\right),i=0,1$ are as defined in (Equation 11). In this context, $Q_{X^{K}}^{\left(0\right)}$ and $Q_{X^{K}}^{\left(1\right)}$ in (Equation 21) are the I-projections [27] of $P_{X^{K}}^{\left(0\right)}$ and $P_{X^{K}}^{\left(1\right)}$ onto this linear family, respectively, which also induces the two exponential families $E_{X}\left(h^{*},P_{X^{K}}^{\left(0\right)}\right)$ and $E_{X}\left(h^{*},P_{X^{K}}^{\left(1\right)}\right)$ with $h^{*}$ as their common natural statistic. Additionally, all the points in $D_{0}\left(E^{*}\right)$ and $D_{1}\left(E^{*}\right)$ are divided by the the linear family $L_{X}\left(h^{*}\right)$.

### 4.3. Local Information Geometric Analysis

Although an explicit information geometry has been shown, we apply the local information geometric framework [13] to provide fundamental insights into this problem. Some useful notations and definitions in local information geometry are introduced as follows.

**Definition** **4**(ϵ-neighborhood)**.**
*Given a finite alphabet Z, and letting $R_{Z}$ be a distribution supported on Z with all entries being positive, its ϵ-neighborhood $N_{\epsilon }^{Z}\left(R_{Z}\right)$ is defined as*
$$\begin{array}{c} N_{\epsilon }^{Z}\left(R_{Z}\right)≜\left\{P_{Z}\in P_{Z}:\sum_{z\in Z}\frac{{\left(P_{Z}\left(z\right)-R_{Z}\left(z\right)\right)}^{2}}{R_{Z}\left(z\right)}\leq \epsilon^{2}\right\}. \end{array}$$

Then, with $R_{Z}$ used as the reference distribution, each distribution $P_{Z}\in P_{Z}$ can be equivalently expressed as a vector $\phi \in R^{\left|Z\right|}$ or a function f:Z→R with
(23)$$\begin{array}{c} \phi \left(z\right)≜\frac{P_{Z}\left(z\right)-R_{Z}\left(z\right)}{\sqrt{R_{Z}\left(z\right)}},\;f\left(z\right)≜\frac{\phi (z)}{\sqrt{R_{Z}\left(z\right)}},\;\forall \;z\in Z, \end{array}$$
referred to as the *information vector* and *feature function* associated with $P_{Z}$, respectively. This provides a three way correspondence $P_{Z}↔\phi ↔f$, which will be useful in our derivations.
Based on Definition 4, we introduce the local assumption that
(24)$$\begin{array}{c} P_{X^{k}}^{\left(i\right)}\in N_{\epsilon }^{X}\left(P_{X^{k}}\right),\;\;\mathrm{for}\;\;i=0,1, \end{array}$$
We use $\psi^{\left(i\right)}↔P_{X^{K}}^{\left(i\right)},i=0,1$ to represent the corresponding information vectors [cf. (Equation 23)]. For each k=1,⋯,K, and given feature $f_{k}:X_{k}\to R$, we define the corresponding information vector $\phi_{k}\in R^{|X_{k}|}$, where $P_{X_{k}}≜{\left[P_{X^{K}}\right]}_{X_{k}}$ is used as the reference distribution. Note that for i=0,1, the correspondence $B_{k}^{T}\psi^{\left(i\right)}↔P_{X_{k}}^{\left(i\right)}$ exists, where $P_{X_{k}}^{\left(i\right)}≜{\left[P_{X^{K}}^{\left(i\right)}\right]}_{X_{k}}$ represents the corresponding marginal distributions. Specifically, $B_{k}$ is an $\left|X|\times \right|X_{k}|$ dimensional matrix with entries [29]
(25)$$\begin{array}{c} B_{k}\left(x^{K},{\hat{x}}_{k}\right)≜\sqrt{\frac{P_{X^{K}}\left(x^{K}\right)}{P_{X_{k}}\left({\hat{x}}_{k}\right)}}\delta_{x_{k}{\hat{x}}_{k}}, \end{array}$$
where $\delta_{x_{k}{\hat{x}}_{k}}$ represents the Kronecker delta.

Moreover, the feature $f_{k}$ defined on $X_{k}$, when considered as a mapping from X to R, corresponds to the information vector $B_{k}\phi_{k}$ in $R^{\left|X\right|}$. Leveraging this correspondence, we can further establish the information vector for $h\left(x^{K}\right)=\sum_{k=1}^{K}f_{k}\left(x_{k}\right)$ as
(26)$$\begin{array}{c} \sum_{i=1}^{K}B_{i}\phi_{i}=B_{0}\phi_{0}\in R^{\left|X\right|}, \end{array}$$
where we have defined
(27)$$\begin{array}{c} B_{0}≜\left[\begin{array}{ccc} B_{1} & \cdots & B_{K} \end{array}\right]\;\mathrm{and}\;\phi_{0}≜\left[\begin{array}{c} \phi_{1}\\ \vdots \\ \phi_{K} \end{array}\right], \end{array}$$
and where for each k=1,⋯,K, $\phi_{k}\in R^{|X_{k}|}$ denotes the information vector corresponding to $f_{k}$.

Additionally, given a matrix $A\in R^{m_{1}\times m_{2}}$, we use $A^{†}$ to denote its Moore–Penrose inverse [30], and we define the associated column space $R\left(A\right)≜\left\{Ax:x\in R^{m_{2}}\right\}$ and projection matrix $\Pi_{A}≜AA^{†}$. Then, we can establish the local counterpart of $E^{*}$ in Theorem 1 as follows.

**Theorem** **2.**
*Under the local assumption *(24)*, let $\psi^{\left(i\right)}↔P_{X^{K}}^{\left(i\right)}$, i=0,1 denote the corresponding information vectors. Then, for $h^{*}$ as defined in *(17)*, we have the correspondence $h^{*}↔B_{0}\phi_{0}^{*}$, where*

(28)
$$\begin{array}{c} \phi_{0}^{*}≜B_{0}^{†}\left(\psi^{\left(1\right)}-\psi^{\left(0\right)}\right), \end{array}$$

*and where $B_{0}$ is defined in *(27)*. In addition, the optimal exponent $E^{*}$ in *(10)* can be expressed as*

(29)
$$\begin{array}{c} E^{*}=\frac{1}{8}{∥B_{0}\phi_{0}^{*}∥}^{2}+o\left(\epsilon^{2}\right). \end{array}$$



**Proof.** See Appendix D. □

Note that from Theorem 2, we have
$$\begin{array}{c} h^{*}↔B_{0}B_{0}^{†}\left(\psi^{\left(1\right)}-\psi^{\left(0\right)}\right)=\Pi_{B_{0}}\left(\psi^{\left(1\right)}-\psi^{\left(0\right)}\right), \end{array}$$
where $\Pi_{B_{0}}$ is the projection matrix associated with the subspace $R(B_{0})$. The optimal feature $B_{0}\phi_{0}^{*}$ in (Equation 26) corresponds to the projection of the sufficient statistic $f_{\mathrm{LLR}}↔\left(\psi^{\left(1\right)}-\psi^{\left(0\right)}\right)$ onto the function space that encompasses all possible *h*’s satisfying the form $h\left(x^{K}\right)=\sum_{k=1}^{K}f_{k}\left(x_{k}\right)$. In other words, $B_{0}\phi_{0}^{*}$ represents the best approximation of $f_{\mathrm{LLR}}$ within the function space of interest, which leads to the optimal decision error exponent $E^{*}$ as shown in (Equation 29).

Moreover, from (Equation 26), this optimal feature can be decomposed to *K* components in subspaces $R(B_{k})$, for k=1,⋯,K,
(30)$$\begin{array}{c} B_{0}\phi_{0}^{*}=\sum_{k=1}^{K}B_{k}\phi_{k}^{*}, \end{array}$$
where $\phi_{0}^{*}$ is stacked by $\phi_{k}^{*}\in R^{|X_{k}|}$, k=1,⋯,K, as in (Equation 27). This decomposition structure can be depicted as Figure 4 for the case K=2.

**Remark** **1.**
*The vectors $B_{i}\phi_{k}^{*}$ are not simply the orthogonal projections of $B_{0}\phi_{0}^{*}$ onto the subspaces $R(B_{k})$ since these subspaces, for k=1,…,K, are not mutually orthogonal. Therefore, the decomposition of $B_{0}\phi_{0}^{*}$ will depend on the Gram matrix [30] of the subspaces $R(B_{k})$, as illustrated in Figure 4. Furthermore, it is noteworthy that the orthogonal projection of $B_{0}\phi_{0}^{*}$ onto the subspaces $R(B_{k})$ can be interpreted as characterizing the optimal error exponent of the binary hypothesis testing problem solely with the observations of $X_{k}$ [12]. When the subspaces $R(B_{k})$ are orthogonal to each other, the optimal inference approach is straightforward, involving the extraction of the optimal information from each node by orthogonal projection. However, when the subspaces $R(B_{k})$ are not orthogonal, different nodes may share various forms of common information. Our result fundamentally demonstrates how to handle this shared information and extract the optimal features through the decomposition of the information vector over non-orthogonal subspaces. This insight provides a novel approach to address the challenges posed by the non-orthogonal subspaces and reveals how to extract the most informative features effectively, ultimately leading to improved performance in the distributed hypothesis testing problem.*


## 5. Type-Based Hypothesis Testing over AWGN Channels

This section presents the optimal error exponent of the type-based hypothesis testing problem over AWGN channels, along with the corresponding coding strategy. To begin, we introduce several notations that will help in the presentation of the results.

**Definition** **5.**
*Let [K]≜{1,2,⋯,K}, and for subset ω⊆[K], i=0,1, we define*

(31)
$$\begin{array}{c} D_{i}^{\omega }\left({\left\{R_{X_{k}}\right\}}_{k\in \omega }\right)≜\min_{Q_{X^{K}}\in S_{\omega }}\;D\left(Q_{X^{K}}∥P_{X^{K}}^{\left(i\right)}\right), \end{array}$$

*where*

$$\begin{array}{cc} S_{\omega }≜ & \left\{Q_{X^{K}}:{\left[Q_{X^{K}}\right]}_{X_{k}}=R_{X_{k}},\;k\in \omega \right\}. \end{array}$$



It would be easy to find that $D_{i}^{\left[K\right]}\left(\cdot \right)=D_{i}^{*}\left(\cdot \right)$, and $D_{i}^{*}\left(\cdot \right)$ is as defined in (Equation 9). Moreover, we define the following error exponent with respect to ω⊆[K].
(32)$$\begin{array}{cc} E_{\omega }≜\min_{{\left\{R_{X_{k}}\right\}}_{k\in \omega },{\left\{\theta_{k}\right\}}_{k\in [K]\setminus \omega }}\;\max\{ & D_{0}^{\omega }\left({\left\{R_{X_{k}}\right\}}_{k\in \omega }\right)+\sum_{k\in [K]\setminus \omega }\frac{{\left(\theta_{k}-\sqrt{p_{k}}\right)}^{2}}{2\mu \sigma_{k}^{2}},\\ \; & D_{1}^{\omega }\left({\left\{R_{X_{k}}\right\}}_{k\in \omega }\right)+\sum_{k\in [K]\setminus \omega }\frac{{\left(\theta_{k}+\sqrt{p_{k}}\right)}^{2}}{2\mu \sigma_{k}^{2}}\}, \end{array}$$
where we have used A∖B to represent the relative complement of set *B* in set *A*, and where μ is as defined in (Equation 8). We can also find $E_{\left[K\right]}=E^{*}$ and $E^{*}$ is as defined in (Equation 10). Finally, we define the quantity $E^{⊙}$, which will be shown as the optimal error exponent
(33)$$\begin{array}{c} E^{⊙}≜\min_{\omega \in ℑ([K])}E_{\omega }, \end{array}$$
where ℑ([K]) denotes the power set of [K].

**Theorem** **3.**
*The optimal error exponent of *(4)* is given by*

(34)
$$\begin{array}{c} \lim_{n\to \infty }-\frac{1}{n}\log P_{n}\left(\hat{H}\neq H\right)=E^{⊙}. \end{array}$$



In the following, we prove Theorem 3 by both the achievability and converse result.

### 5.1. The Coding Strategy for Distributed Nodes

First, we define the different regimes of empirical distributions, for each k=1,⋯,K and for some γ∈(0,1). Basically, the specific choice of γ does not effect the achievable error exponent as long as γ∈(0,1). It helps conduct the decode-and-forward and amplify-and-forward coding strategies as introduced in Section 1.



**Decode-and-forward regime:**


$$\begin{array}{cc} \; & M_{k}^{\left(0\right)}≜\left\{R_{X_{k}}:D\left(R_{X_{k}}∥P_{X_{k}}^{\left(0\right)}\right)<n^{-\gamma }\right\},\\ \; & M_{k}^{\left(1\right)}≜\left\{R_{X_{k}}:D\left(R_{X_{k}}∥P_{X_{k}}^{\left(1\right)}\right)<n^{-\gamma }\right\}. \end{array}$$





**Amplify-and-forward regime:**


(35)
$$\begin{array}{c} M_{k}^{c}≜\left\{R_{X_{k}}:\min\left\{D\left(R_{X_{k}}∥P_{X_{k}}^{\left(0\right)}\right),D\left(R_{X_{k}}∥P_{X_{k}}^{\left(1\right)}\right)\right\}\geq n^{-\gamma }\right\}. \end{array}$$



Note that for each k=1,⋯,K, the probability of the empirical distribution ${\hat{P}}_{X_{k}}$ in $M_{k}^{c}$ is $\exp\left(-n^{1-\gamma }\right)$. Consequently, in the amplify-and-forward regime, we can transmit such empirical distributions with exponentially large power by Pulse Amplitude Modulation (PAM) while still satisfying the power constraint. Specifically, let $P_{X_{k}}^{\left(n\right)}$ be the set of all possible empirical distributions of $X_{k}$ with *n* samples, and denote $\eta_{k}≜\left|P_{X_{k}}^{\left(n\right)}\cap M_{k}^{c}\right|$. We define the bijective function $\xi_{k}:P_{X_{k}}^{\left(n\right)}\cap M_{k}^{c}\mapsto \left\{1,\ldots ,\eta_{k}\right\}$ as the indices of empirical distributions. Then, according to the observed empirical distribution, the encoder of node *k* (k=1,⋯,K) is designed to transmit the signal
(36)$$\begin{array}{c} Q_{k}\left({\hat{P}}_{X_{k}}\right)≜\xi_{k}\left({\hat{P}}_{X}\right)\cdot \exp\left(n^{\frac{1-\gamma }{2}}\right). \end{array}$$

Furthermore, if the empirical distributions are in the decode-and-forward regimes, we initially detect the true hypothesis and then transmit the bit using Binary Phase Shift Keying (BPSK) with the appropriate power. By employing these strategies, the achievability result can be obtained through repeated transmissions from all the distributed nodes. In other words, the resulting encoder for node *k* is defined as follows:(37)$$\begin{array}{c} g_{k}^{*}=\left[g_{k}^{*},\cdots ,g_{k}^{*}\right],\;k=1,\cdots ,K, \end{array}$$
where
(38)$$\begin{array}{c} g_{k}^{*}\left({\hat{P}}_{X_{k}}\right)≜\left\{\begin{array}{cc} \sqrt{p_{k}-\delta \left(n,\gamma \right)}, & \mathrm{if}\;{\hat{P}}_{X_{k}}\in M_{k}^{\left(0\right)}\\ -\sqrt{p_{k}-\delta \left(n,\gamma \right)}, & \mathrm{if}\;{\hat{P}}_{X_{k}}\in M_{k}^{\left(1\right)}\\ Q_{k}\left({\hat{P}}_{X_{k}}\right), & \mathrm{if}\;{\hat{P}}_{X_{k}}\in M_{k}^{c} \end{array}\right., \end{array}$$
and where
(39)$$\begin{array}{c} \delta \left(n,\gamma \right)≜\max_{k\in [K]}\;\frac{P_{n}\left({\hat{P}}_{X_{k}}\in M_{k}^{c}\right)}{P_{n}\left({\hat{P}}_{X_{k}}\notin M_{k}^{c}\right)}\cdot {\left(n+1\right)}^{2|X_{k}|}\cdot \exp\left(2n^{\frac{1-\gamma }{2}}\right). \end{array}$$

**Proposition** **2.**
*The encoders as defined in *(38)* satisfy the power constraint *(6)*, and*

(40)
$$\begin{array}{c} \lim_{n\to \infty }\delta \left(n,\gamma \right)=0. \end{array}$$



**Proof.** See Appendix E. □

### 5.2. Decision Rule and Achievable Error Exponent

After the decision center receives the output signals $g_{1}^{*}\left({\hat{P}}_{X_{1}}\right)+Z_{1},\cdots ,g_{K}^{*}\left({\hat{P}}_{X_{K}}\right)+Z_{K}$, we then compute
$$\begin{array}{c} \theta_{k}≜\frac{1}{m}\sum_{i=1}^{m}{\left[g_{k}^{*}\left({\hat{P}}_{X_{k}}\right)+Z_{k}\right]}_{i},\;k=1,\cdots ,K, \end{array}$$
where ${\left[\cdot \right]}_{i}$ denotes the *i*-th entry of a given vector. Then, we conduct the log-likelihood ratio test (LLRT) to detect the true hypothesis:(41)$$\begin{array}{c} \log\frac{P_{n}\left(\theta_{1},\cdots ,\theta_{K}|H=H_{0}\right)}{P_{n}\left(\theta_{1},\cdots ,\theta_{K}|H=H_{1}\right)}\overset{\hat{H}=H_{0}}{\underset{\hat{H}=H_{1}}{⋛}}0. \end{array}$$

Note that exponentially large power is allocated for the empirical distributions in the amplify-and-forward regime (cf. (Equation 35), (Equation 36)); the decision center can correctly detect the coding regime of the nodes with super-exponentially high probability, i.e., for k=1,⋯,K,
(42)$$\begin{array}{cc} \; & \lim_{n\to \infty }-\frac{1}{n}\log P_{n}\left({\hat{P}}_{X_{k}}\in M_{k}^{c}|\theta_{k}\leq \exp\left(n^{\frac{1-\gamma }{4}}\right)\right)=\infty ,\\ \; & \lim_{n\to \infty }-\frac{1}{n}\log P_{n}\left({\hat{P}}_{X_{k}}\notin M_{k}^{c}|\theta_{k}>\exp\left(n^{\frac{1-\gamma }{4}}\right)\right)=\infty . \end{array}$$
Therefore, we can assume that the decision center knows the coding regime of the nodes and define the following regime of the received signals with respect to subset ω⊆[K].
$$\begin{array}{cc} \; & \Theta_{\omega }≜\left\{\left(\theta_{1},\cdots ,\theta_{K}\right):\theta_{k}>\exp\left(n^{\frac{1-\gamma }{4}}\right),\forall k\in \omega ,\;\mathrm{and}\;\theta_{k^{\prime }}\leq \exp\left(n^{\frac{1-\gamma }{4}}\right),\forall k^{\prime }\in \left[K\right]\setminus \omega \right\}, \end{array}$$
for all ω∈ℑ([K]). When the received signals $\left(\theta_{1},\cdots ,\theta_{K}\right)\in \Theta_{\omega }$, the decision center can recover the empirical distributions ${\hat{P}}_{X_{k}}$ (k∈ω) from the received signals $\theta_{k}$ by the decoder:(43)$$\begin{array}{cc} \; & Q_{k}^{-1}\left(\theta_{k}\right)≜\xi_{k}^{-1}\left(\left\lfloor \theta_{k}/\exp\left(n^{\frac{1-\gamma }{2}}\right)+0.5\right\rfloor \right), \end{array}$$
where ⌊·⌋ denotes the floor function [31]. The following result shows that decoding error of (Equation 43) can be neglected.

**Proposition** **3.**
*For all ${\hat{P}}_{X_{k}}\in P_{X_{k}}^{\left(n\right)}\cup M_{k}^{c}$, k=1,⋯,K,*

(44)
$$\begin{array}{c} \lim_{n\to \infty }-\frac{1}{n}\log P\left(Q_{k}^{-1}\left(\theta_{k}\right)\neq {\hat{P}}_{X_{k}}\right)=\infty . \end{array}$$



**Proof.** See Appendix F. □

In the following, we denote $p_{k}^{\prime }≜p_{k}-\delta$, for k=1,⋯,K and discuss the decision error exponent when the received signals are in $\Theta_{\omega }$. For k∈ω, the empirical distribution ${\hat{P}}_{X_{k}}$ can be recovered by (Equation 43), and for k∈[K]∖ω, node *k* detects the hypothesis according to the observed empirical distribution and transmits the detected bit by BPSK (cf. (Equation 38)) through the AWGN channel. Then, the decision center detects the true hypothesis from the received signals by LLRT (Equation 41), which can be reduced to
(45)$$\begin{array}{c} {\tilde{E}}_{0}^{\omega }\left(\theta_{1},\cdots ,\theta_{K}\right)\overset{\hat{H}=H_{1}}{\underset{\hat{H}=H_{0}}{⋛}}{\tilde{E}}_{1}^{\omega }\left(\theta_{1},\cdots ,\theta_{K}\right), \end{array}$$
where for i=0,1,
$$\begin{array}{cc} \; & {\tilde{E}}_{i}^{\omega }\left(\theta_{1},\cdots ,\theta_{K}\right)\\ \; & ≜\min_{\bar{\omega }\in ℑ\left(\left[K\right]\setminus \omega \right)}\;D_{i}^{*}\left({\bar{P}}_{X_{1}},\cdots ,{\bar{P}}_{X_{K}}\right)+\sum_{k\in \bar{\omega }}\frac{{\left(\theta_{k}-\sqrt{p_{k}^{\prime }}\right)}^{2}}{2\mu \sigma_{k}^{2}}+\sum_{k^{\prime }\in \left[K\right]\setminus \left(\omega \cup \bar{\omega }\right)}\frac{{\left(\theta_{k^{\prime }}+\sqrt{p_{k^{\prime }}^{\prime }}\right)}^{2}}{2\mu \sigma_{k^{\prime }}^{2}}, \end{array}$$
where ℑ([K]∖ω) denotes the power set of [K]∖ω, and where for k=1,⋯,K,
(46)$$\begin{array}{c} {\bar{P}}_{X_{k}}≜\left\{\begin{array}{cc} Q_{k}^{-1}\left(\theta_{k}\right),\; & \mathrm{if}\;k\in \omega \\ \;P_{X_{k}}^{\left(0\right)},\; & \mathrm{if}\;k\in \bar{\omega }\\ \;P_{X_{k}}^{\left(1\right)},\; & \mathrm{if}\;k\in \left[K\right]\setminus \left(\omega \cup \bar{\omega }\right) \end{array}\right.. \end{array}$$
Consequently, the decision error exponent is characterized by the following proposition.

**Proposition** **4.**
*For any ϵ>0 and ω∈ℑ([K]), the decision error exponent by the decision rule *(45)* satisfies*

(47)
$$\begin{array}{c} \lim_{n\to \infty }-\frac{1}{n}\log P_{n}\left(\hat{H}\neq H,\left(\theta_{1},\cdots ,\theta_{K}\right)\in \Theta_{\omega }\right)\geq E^{⊙}-\epsilon , \end{array}$$

*where $E^{⊙}$ is as defined in *(33)*.*


**Proof.** See Appendix G. □

Noticing that the overall decision error probability is
$$\begin{array}{c} P_{n}\left(\hat{H}\neq H\right)=\sum_{\omega \in ℑ([K])}P\left(\hat{H}\neq H,\left(\theta_{1},\cdots ,\theta_{K}\right)\in \Theta_{\omega }\right), \end{array}$$
the following proposition establishes the achievable error exponent by the coding strategy (Equation 38).

**Proposition** **5.**
*By using the encoders $g_{1}^{*},\cdots ,g_{K}^{*}$ as defined in *(38)*, and the decision rules $\hat{H}$ from *(41)*, the achievable error exponent is given by $E^{⊙}$, i.e.,*

(48)
$$\begin{array}{c} \lim_{n\to \infty }-\frac{1}{n}\log P_{n}\left(\hat{H}\neq H\right)\geq E^{⊙}, \end{array}$$

*where $E^{⊙}$ is as defined in *(33)*.*


### 5.3. The Converse Result

In this section, we show that $E^{⊙}$ is indeed an upper bound of (Equation 4), which establishes Theorem 3. Our main technique is to apply a genie-aided approach, which provides different kinds of additional information to both nodes and computes the corresponding error exponents under additional information. As depicted in Figure 5, given index set ω∈ℑ([K]), suppose that for all k∈ω, node *k* can know and cancel the channel noise in advance; then, the channel is noiseless, and the decision center can perfectly receive the empirical distribution ${\hat{P}}_{X_{k}}$. On the other hand, suppose that for all $k^{\prime }\in \left[K\right]\setminus \omega$, we can leverage the true hypothesis H to node $k^{\prime }$; then, with such additional information, we can establish the following upper bound of (Equation 4) (cf. (Equation 33)).

**Proposition** **6.**
*Given index set ω∈ℑ([K]), suppose that for all k∈ω, the decision center can obtain ${\hat{P}}_{X_{k}}$ perfectly. Additionally, for all $k^{\prime }\in \left[K\right]\setminus \omega$, node $k^{\prime }$ can obtain the true hypothesis H. The resulting optimal decision error exponent is*

(49)
$$\begin{array}{c} \lim_{n\to \infty }-\frac{1}{n}\log P_{n}\left(\hat{H}\neq H\right)=E_{\omega }, \end{array}$$

*where $E_{\omega }$ is as defined in *(32)*.*


**Proof.** See Appendix H. □

Notice that Proposition 6 is verified for all ω∈ℑ([K]), and we cannot obtain a better performance than Proposition 6 for the DHT over AWGN channels without the additional information. We then conclude the following error exponent upper bound.

**Proposition** **7.**
*For all possible encodes $g_{1},\cdots ,g_{K}$ under the power constraint *(6)*, the corresponding error exponent with respect to the LLRT decision rule satisfies*

(50)
$$\begin{array}{c} \lim_{n\to \infty }-\frac{1}{n}\log P_{n}\left(\hat{H}\neq H\right)\leq E^{⊙}, \end{array}$$

*where $E^{⊙}$ is as defined in *(33)*.*


Finally, by combining Propositions 5 and 7, Theorem 3 is proved.

**Remark** **2**(Local-geometric interpretation)**.**
*Note that the expression of the optimal error exponent $E^{⊙}$ as defined in *(Equation 33)* is quite intricate, which could limit our understanding. To simplify the analysis, we introduce the local geometry assumption as given in *(Equation 24)*. In Appendix I, we demonstrate that the error exponent corresponds to a more manageable expression*
(51)$$\begin{array}{c} E^{⊙}=\min_{\omega \in ℑ([K])}\;\frac{1}{8}{∥B_{\omega }B_{\omega }^{†}\left(\psi_{\omega }^{\left(1\right)}-\psi_{\omega }^{\left(0\right)}\right)∥}^{2}+\sum_{k\in [K]\setminus \omega }\frac{p_{k}}{2\mu \sigma_{k}^{2}}+o\left(\epsilon^{2}\right), \end{array}$$
*where for $\omega =\{i_{1},\cdots ,i_{\left|\omega \right|}\}$, we have defined*
(52)$$\begin{array}{c} B_{\omega }≜\left[\begin{array}{ccc} B_{i_{1}} & \cdots & B_{i_{\left|\omega \right|}} \end{array}\right], \end{array}$$
*and $\psi_{\omega }^{\left(i\right)}↔{\left[P_{X^{K}}^{\left(i\right)}\right]}_{X_{i_{1}}\cdots X_{i_{\left|\omega \right|}}},i=0,1$. Given ω∈ℑ([K]), the first term in *(Equation 51)* represents the optimal error exponent (cf. *(Equation 29)*) when the decision center can access the empirical distributions ${\hat{P}}_{X_{k}}$, k∈ω. The second term corresponds to the optimal error exponent when node k, k∈[K]∖ω can know the true hypothesis H and transmit the bit using BPSK modulation. The total error exponent is the sum of these two parts, and $E^{⊙}$ aims to determine the minimum sum among all possible splits of the index set [K]. In other words, $E^{⊙}$ finds the optimal trade-off between accessing empirical distributions at the decision center and having individual nodes transmit bits with BPSK modulation.*

## 6. Discussion

This paper discusses the DHT problem over two communication models.The first is the noiseless channel, which is mostly considered in current distributed learning and federated learning systems [9,11]. For the noiseless channels, we show that by using one-dimensional statistics from different nodes, it is possible to achieve the same error exponent when the decision center has knowledge of the corresponding empirical distributions. This result is significant as it simplifies the coding process at distributed nodes, allowing them to transmit only the necessary statistics rather than the entire empirical distribution, which provides a practical implementation of the result in [5]. This finding proves the rationality of transmitting statistics as the most widely-used strategy in distributed learning and federated learning [11].

For the AWGN channels, this paper introduces a novel coding strategy, which cleverly combines decode-and-forward and amplify-and-forward techniques. The underlying concept of this coding strategy is based on the observation that the probability of the empirical distribution deviating significantly from the true marginal distribution diminishes exponentially. Consequently, by employing sufficiently large power, we can transmit the empirical distribution almost perfectly to the decision center while satisfying the averaged power constraint. When the prior distributions are not 1/2, the strategy still work for the optimal error exponent, and the only difference is to adjust the BPSK points for two hypotheses according to the power constaint. The demonstrated optimality of the achieved decision error exponent further indicates that the proposed coding strategy is highly effective and successfully approaches the theoretical limit within the given constraints of the problem.

## 7. Conclusions

This paper focuses on investigating DHT problems over both noiseless channels and AWGN channels, where the distributed nodes are constrained to encoding the received empirical distributions, driven by practical computational considerations. In the first problem, we demonstrate that utilizing one-dimensional statistics of distributed nodes and simply summing them up as the decision rule can lead to the optimal error exponent. For the second problem, we propose a coding strategy that combines decode-and-forward and amplify-and-forward techniques. We further introduce a genie-aided approach to establish the optimality of the achieved decision error exponent. Overall, our findings offer valuable insights into coding techniques for distributed nodes, and the established strategies can be extended to more general scenarios, broadening the applicability of DHT in diverse settings.

## Figures and Tables

**Figure 1 entropy-25-01434-f001:**
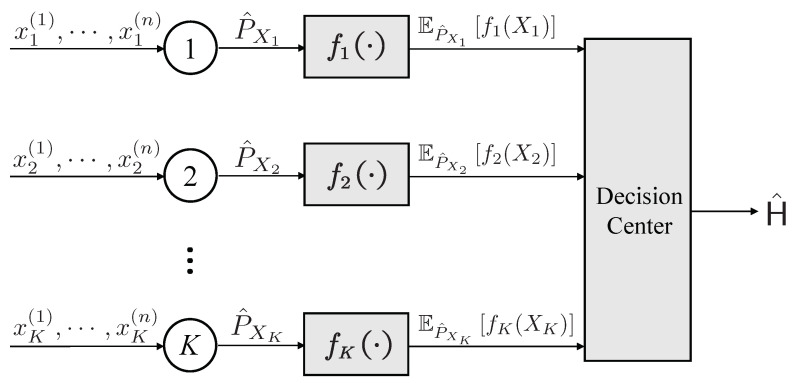
The transmission procedures for the type-based distributed hypothesis testing problem over noiseless channels.

**Figure 2 entropy-25-01434-f002:**
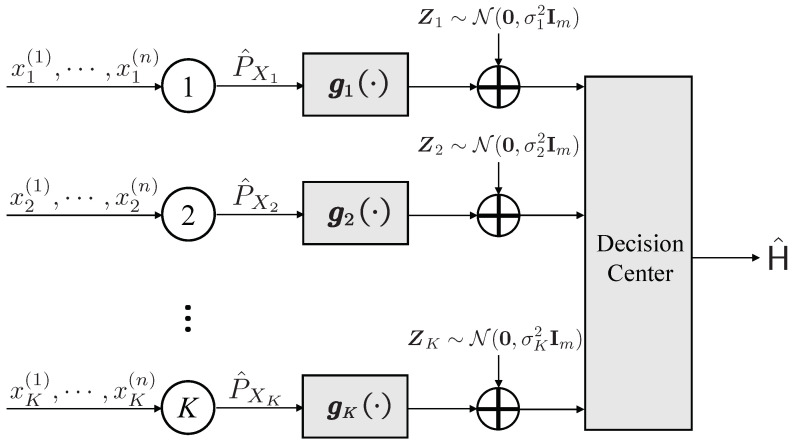
The transmission procedures for the type-based distributed hypothesis testing problem over AWGN channels.

**Figure 3 entropy-25-01434-f003:**
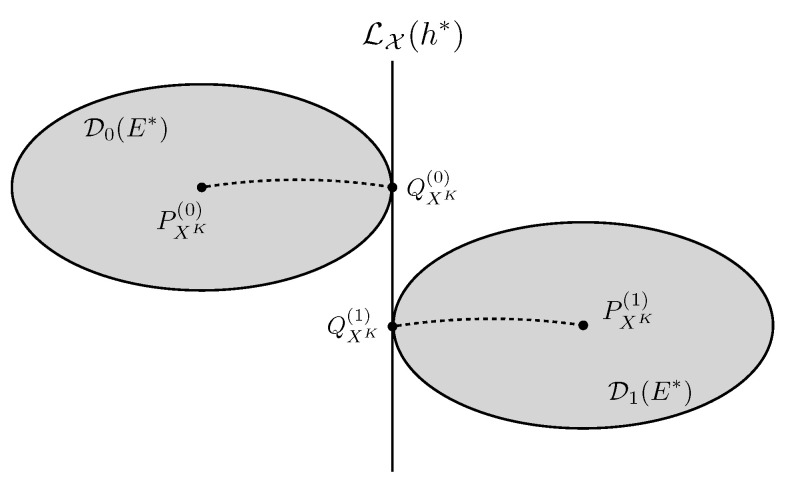
The geometric structure in distributed hypothesis testing, with $Q_{X^{K}}^{\left(i\right)}$ denoting the I-projection of $P_{X^{K}}^{\left(i\right)}$ onto the linear family $L_{X}\left(h^{*}\right)$, i=0,1, and $L_{X}\left(h^{*}\right)$ can devide $D_{0}\left(E^{*}\right)$ and $D_{1}\left(E^{*}\right)$ in different half spaces.

**Figure 4 entropy-25-01434-f004:**
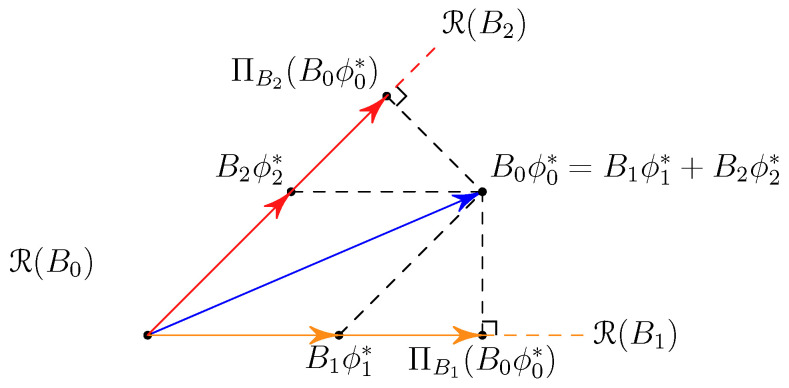
The information decomposition structure in distributed hypothesis testing with K=2 nodes, compared with the orthogonal decompositions on the subspace $R(B_{k})$ for each node k=1,2.

**Figure 5 entropy-25-01434-f005:**
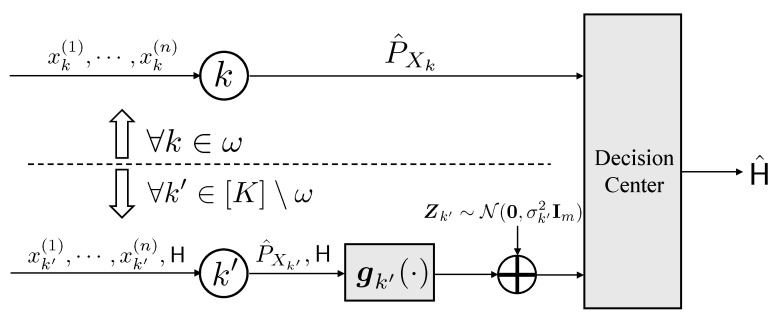
A geometric explanation of the genie-aided approach, which can lead to $E_{\omega }$ as the upper bound of the error exponent in (Equation 4).

## Data Availability

Data sharing not applicable.

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
