# Peer review of "On the Optimal Error Exponent of Type-Based Distributed Hypothesis Testing†"

_entropy, 2023, doi:10.3390/e25101434_

Round 1

Reviewer 1 Report

It is rather ironic that in the first sentence of the abstract, the authors assert that with many works in the field, "the 1 information-theoretic optimality of coding strategies is often inadequately addressed". While they do not provide evidence for this statement, this describes very accurately the work at hand.

The paper considers two scenarios, let me concentrate on the first. The terminals observe sequences of length n. They are allowed to send the mean of some scalar function of the observations, and the exponential decay of the decision error probability (w.r.t. a uniform prior on the observations) is evaluated. I see at least three problems with this.

1. Lack of limitation on communications. The terminals are allowed to send a scalar, at an arbitrarily high resolution. Only the observation is limited, communication isn't. To avoid sending e.g. the whole observed sequence, a computational constraint is imposed, and only one mean may be evaluated. This is very awkward. Especially in an information-theoretic setting, how can you ignore the number of bits being sent? And in a practical setting, why is the computational constraint in the form of evaluating a single mean? You don't worry about the number of elements that you average (which goes to infinity), you only care about the number of averages that you take? Such choices are very strange, and require serious justification.

2. Limitation on the distributions. If the marginal distribution at some terminal is the same under both hypotheses, then sending any one-dimensional statistic will result in a non-decaying error probability. The reason is, that under the law of large numbers this statistic will concentrate to the expected value, which is hypothesis-invariant. Thus, the work at hand is only relevant to scenarios where the marginals depend on the hypothesis. Such scenarios are of less interest in the distributed setting as even a single terminal can achieve exponential decay, so if that single terminal also performs the test - no communication at all is needed. Indeed, for this reason some recent works limit themselves to the case where the marginals are the same and it is the correlation that needs to be tested. Of course, the authors may argue differently, but they must at least address the issue.

3. Choice of error criterion. The work evaluates the exponent under a uniform prior. This equals of course the exponent under any non-trivial prior. It is the point on the exponent tradeoff, where the type-I and type-II exponents are balanced. Why was this point chosen, rather than any other point on the tradeoff? In the exponential regime, this is a completely arbitrary choice, and no explanation is given.

To summarize, I cannot recommend to publish this work. If the authors wish it to be seriously considered, they must explain all of the choices they make, that differ from more standard analysis. Each of these choices should be justified, and then the effect of this choice should be analyzed separately.

Can be improved, but nothing major.

Author Response

For the abstract “the information-theoretic optimality of coding strategies is often inadequately addressed”, the explanation is just in the first paragraph of introduction, the results of both achievability and converse are established in limited cases. To address the reviewer’s concern, we have modified it.

  1. (Lack of limitation on communications) We thank the reviewer’s suggestion. Our work is partly motivated by practical federated learning. In such cases, doing empirical mean is the most common strategy, while making other coding strategies will face some problems. For instance, when the data provided are in different batches, it would be difficult to encode the empirical distribution but easy to compute the empirical means (You can just collect the empirical means of different batches and compute their empirical mean). For the setting of noiseless channels, it considers the real applications in federated learning, where usually the transmission between two computers throughout the internet can achieve enough precision, although it may mean a large bit number and need a certain detailed mechanism. Actually, the optimal exponent is just the same as transmitting the empirical distributions, what we have done is that we improve the efficiency by better design, since computing empirical means is the most convenient way in federated learning or distributed learning. To address the reviewer’s concern, we add some explanations in the setting of noiseless channels.
  2. (Limitation on the distributions.) The theme of our paper is “type-based”, which is motivated by practical federated learning applications. As the reviewer has pointed out, with only empirical distributions, hypothesis testing against independence, i.e., two hypotheses share the same marginal distributions, can not be accomplished. The reviewer pointed out that the case we have set is of less interest, which is mostly for the typical distributed hypothesis testing works. However, for practical federated learning, it will be impossible that the data distributions are the same for different labels. Although a single terminal can achieve exponential decay, the correlation in multiple terminals can improve the error exponent, which is meaningful, since we will not abandon using the data from certain terminals in real applications. To address the reviewer’s concern, we have added some explanations when providing such settings. 
  3. (Choice of error criterion) For the result of noiseless channels, the optimal error exponent and decision rule would not change for any given prior distribution. As the reviewer pointed out, this is a completely arbitrary choice. For the result of AWGN channels, the optimal error exponent will depend on the prior distribution, say, P(H_0)=\alpha and P(H_1)=1-\alpha. The reason is that the BPSK strategy will adjust accordingly to utilize the power constraint. However, the mixture coding strategy still works. We do not present the result under given \alpha because the current result is already complicated in math, importing \alpha would aggravate the complexity. We thus use the setting of equal priors, and in view of the consistency, we take it as a general setting for the whole paper. To address the reviewer’s doubt, we have deleted the prior setting in noiseless channels, and added a comment in the discussion part for AWGN channels.

Reviewer 2 Report

The authors present in this paper the error exponent for two distributed models: the first concerns noiseless channels, and the second deals with the Gaussian ones. At the beginning, I didn't understand why both results were provided in the same paper; but finally, the tools for the proofs were shared.

The notations and the hypothesis part come too late: the hypothesis described in Section 2.2 should be provided along with the problem formulation and not in a separate section. Also, the definition 1 should be moved where it is needed and not at the beginning. Otherwise, the reader is lost between the big ideas and the details that are not yet relevant in the text.

In the formulation of the noiseless channels case, the features $f_k$ are not restricted and thus may be carefully chosen in order to provide the complete type to the decision center! E.g. f(P_k) = 0.005004002 if x_1 is seen 5 times, x_2 4 times and x_3 only twice. n is fixed, this can be made linear as required by the form (3). In our case f(x_1) = 0.001, f(x_2) = 0.000001 and f(x_3) = 000000001.

The dot equal notation deserves an explanation.

Please provide some intuition about the definition. For example, what is the meaning of S in definition 2? It will avoid unnecessary reflection from the reader.

Section 4.1 should be split: first the optimal feature discussion in its own section and then the geometric structure in another one. Moreover, Figure 3 is not clear enough. It would be better if the sets D^*_0 and D^*_1 were plotted with a separating line (keep it 2D).

Is the local information geometric analysis a linearisation of the problem around the separating plane h? The condition (24) seems restrictive as the hypothesis probabilities should be close if (24) must hold for both i=0 and i=1.

Finally, the famous book by Hardy and Wright, reference 32, is cited only to support the floor function. That's an overkill!

Author Response

We thank the reviewer for the valuable advice.

  1. (Content of Section 2.2) We have rearranged the content according to your suggestions.
  2. (The setting of empirical mean) We totally agree the empirical distribution can easily be transmitted to the decision center. The optimal exponent is just the same as transmitting the empirical distributions. What we have done is that we improve the efficiency by better design, since computing empirical means is the most convenient way in federated learning or distributed learning. To address the reviewer’s concern, we add some explanations in the setting of noiseless channels.
  3. (Dot equal notation, explanation of definitions) We have added the explanations accordingly.
  4. (Section 4.1) We have split the main result and geometric explanation, and also modified the figure accordingly
  5. (On the local geometry) The basis of the local geometry actually assumes hypothesis probabilities should be close. Your understanding of linearization is also right. The reason for introducing the local geometry is to avoid the heavy mathematical expressions and show the result with simple analytical expressions, which can provide more insights. 
  6. We also changed the reference 32.

Reviewer 3 Report

This highly theoretical paper is well developed and as such is useful. However, from a point of view of real-world applications, it should include a discussion on the radio propagation of signasl, which is totally missing in this draft. Why, for example, considering an ideal radio channel with no noise? Noise is never absent and its does not casre about mathematical "pictures". I would define "ideal" an AWGN channel, not one with no noise.

What are the conclusions if the channel has Rayleigh (worst case) fading? Which are the carrier frequencies envisaged in the system?

I understand that the paper is very theoretical and that the authors would like to keep it at this level, but, as an engineer of TLC, I would like to read about the framework in which their theory might be applied. 

Author Response

We thank the reviewer for the advice.

  1. (The setting of noiseless channel) We thank the reviewer’s suggestion. Our work is partly motivated by practical federated learning. In such cases, it considers the real federated or distributed applications, where usually the transmission between two computers throughout the internet can achieve enough precision, although it may mean a large bit number and need a certain detailed mechanism, like the setting of carrier frequencies. We hope to emphasize that the ‘noiseless’ is actually common when thinking of deep learning scenarios, and we add some explanation when giving the setting.
  2. (Other channels) To prove the information-theoretical optimality is typically hard. We are happy to consider more possible channels, but actually the coding strategy will change a lot. We would like to study it in the future work.
  3. (Application) Besides math, we hope to provide practical coding strategies for federated learning and distributed learning cases, where we only consider manipulations on the empirical distributions. I think federated learning is the scenario where our result can be applied.

Round 2

Reviewer 3 Report

I would have appreciated some comments on real applications and noisy channles, but for computer people I continue to hear tha noise does not exist.

Author Response

Thanks for your comments, and we would like to add some notes accordingly in the discussion part.